

# Organosulfates in Atlanta, Georgia: Anthropogenic influences on biogenic secondary organic aerosol formation

Anusha Priyadarshani Silva Hettiyadura[1], Ibrahim M. Al-Naiema[1], Dagen D. Hughes[1], Ting Fang[2,3], Elizabeth A. Stone[1]

[1]Department of Chemistry, University of Iowa, Iowa City, IA-52246, USA
[2]School of Earth and Atmospheric Sciences, Georgia Institute of Technology, Atlanta, GA, USA
[3]Now at: University of California, Irvine, Irvine, CA 92697, USA

*Correspondence to*: Elizabeth A. Stone (betsy-stone@uiowa.edu)

**Abstract**

This study examines the anthropogenic influence on biogenic organosulfate formation at an urban site in Atlanta, GA in the Southeastern United States. Organosulfates were analyzed in fine particulate matter ($PM_{2.5}$) collected during August 2015 in Atlanta using hydrophilic interaction liquid chromatography (HILIC), tandem mass spectrometry (MS/MS), and high-resolution time-of-flight (ToF) mass spectrometry. By their MS/MS response, 32 major organosulfate species were identified, selected species were quantified, and other species were semi-quantified using surrogate standards. Organosulfates accounted for 16.5% of $PM_{2.5}$ organic carbon (OC). Isoprene-derived organosulfates were the most abundant, dominated by methyltetrol sulfate which accounted for 12.6% of $PM_{2.5}$ OC. Together, the isoprene-derived organosulfates accounted for the majority of the isoprene-derived secondary organic aerosols (SOA) that had been previously observed in Atlanta, but had not been identified at the molecular level. Other major species included seven monoterpene-derived organosulfates, five diesel and/or biodiesel-derived organosulfates, and three new organosulfates that are also expected to derive from isoprene. Organosulfate species and concentrations in Atlanta were compared to those in a rural forested site in Centreville, AL during summer 2013, which were also dominated by isoprene-derived organosulfates. In Atlanta, isoprene-derived organosulfate concentrations were two to six times higher and accounted for twice as much OC. The greatest enhancement in concentration was observed for 2-methylglyceric acid sulfate, a tracer for isoprene high-$NO_x$ SOA. The isoprene-derived organosulfates indicated a stronger influence of $NO_x$ in Atlanta compared to Centreville. Overall, these results suggest that SOA in the Southeastern US can be reduced by controlling $NO_x$ and $SO_2$ emissions from fossil fuel combustion. This study gives insights into the major organosulfate species that should be targets for future measurements in urban environments and standard development.





# 1 Introduction

Atmospheric organosulfates are components of secondary organic aerosol (SOA) that contain a sulfate ester functional group. Organosulfates have been detected all around the world and are estimated to contribute up to 5-9% of $PM_{2.5}$ OA in the Southeastern US (Tolocka and Turpin, 2012). The organosulfates primarily form by the reactive uptake of
gas-phase epoxides on acidic sulfate particles (Surratt et al., 2010; Lin et al., 2013). Alternatively, they form by the sulfate radical-initiated oxidation of volatile organic compounds (VOC) (Nozière et al., 2010; Schindelka et al., 2013) and nucleophilic substitution of nitrate groups by sulfate (Darer et al., 2011; Hu et al., 2011). Precursors of organosulfates are largely biogenic VOC such as isoprene, monoterpenes, sesquiterpenes, 2-methyl-3-butene-2-ol (MBO), and green leaf volatiles (Zhang et al., 2012; Surratt et al., 2008; Chan et al., 2011; Iinuma et al., 2009; Shalamzari et al., 2014). Since fossil
fuel combustion is the major source of sulfate aerosols in the atmosphere (Wuebbles and Jain, 2001; Hidy et al., 2014; Carlton et al., 2010), organosulfates are tracers of anthropogenically influenced biogenic SOA (Hettiyadura et al., 2018). Organosulfates have also been observed in diesel and biodiesel emissions (Blair et al., 2017) and in SOA produced from anthropogenic VOC (i.e. naphthalene, methylnaphthalene) (Riva et al., 2015) and long chain *n*-alkanes (Riva et al., 2016a), although the significance of these sources to ambient organosulfates has not yet been established.

Atlanta, GA is the 9[th] most populous metropolitan area in the US with a population in 2017 of 5.9 million (U.S. Census Bureau). Here, organic aerosols account for 68-70% of $PM_{2.5}$ during summer (Rattanavaraha et al., 2017; Al-Naiema et al., in preparation) the majority of which is secondary in origin (50-65%) and is strongly influenced by isoprene SOA (Weber et al., 2007). For example, isoprene dihydroxy epoxides (IEPOX) contributed 29-38% of fine OA (Rattanavaraha et al., 2017; Budisulistiorini et al., 2016; Xu et al., 2015a) and total isoprene-derived OA contributed to 27% of $PM_{2.5}$ organic
carbon (OC) (Al-Naiema et al., in preparation). The diurnal variation of IEPOX-OA in Atlanta, GA was temporally consistent with isoprene emissions from plants, suggesting that the IEPOX-OA form locally rather than being transported from surrounding forested sites (Xu et al., 2015b). In Atlanta, sulfate is the second largest component of $PM_{2.5}$ and accounted for 15-21% of $PM_{2.5}$ mass (Rattanavaraha et al., 2017; Al-Naiema et al., in preparation). The aerosol acidity (average pH 1.4 ± 0.7) and the aerosol water content (averaging 8.4 ± 4.8 µg m$^{-3}$) in Atlanta peaks during summer (Rattanavaraha et al.,
2017), similar to other locations in the Southeastern US (Guo et al., 2015). Previous studies have demonstrated that the biogenic SOA formation in the Southeastern US is enhanced by sulfate, $NO_x$, and $O_3$, which are mainly coming from fossil fuel combustion, particularly during summer when the biogenic emissions are high (Goldstein et al., 2009; Gao et al., 2006; Xu et al., 2015a; Carlton et al., 2010).

This study examines the anthropogenic influence on organosulfate formation during summer at an urban site in
Atlanta in the Southeastern US. Our specific objectives include 1) identification and quantification of major organosulfate species in Atlanta, GA during August 2015 using hydrophilic interaction liquid chromatography (HILIC), tandem mass spectrometry (MS/MS), and high-resolution time-of-flight mass spectrometry (ToF-MS), 2) evaluation of the factors that influence organosulfate formation *via* comparison of observed species to SOA chamber experiments and correlations of



organosulfates with SOA tracers, other PM$_{2.5}$ constituents, gas-phase reactive species, and meteorological conditions, and 3) comparison of these results with the major organosulfates identified and quantified in Centreville, AL during summer 2013 (Hettiyadura et al., 2017;  Hettiyadura et al., 2018) to better understand the extent to which anthropogenic pollutants affect biogenic organosulfate formation across an urban and rural pair in the Southeastern US during summer. This study provides insights into the composition, abundance, sources, and formation pathways of organosulfates, which are useful as tracers for anthropogenically-influenced SOA.

## 2 Materials and methods

### 2. 1 Chemicals and reagents

Hydroxyacetone sulfate and glycolic acid sulfate (potassium salts, > 95% purity) were synthesized according to Hettiyadura et al. (2015); lactic acid sulfate (24.9% purity) was synthesized according to Olson et al. (2011); 2-methyltetrol sulfate was synthesized according to Budisulistiorini et al. (2015) and Bondy et al. (2018). Ultra-pure water was prepared on site (Thermo, Barnsted EasyPure-II; 18.2 MΩ-cm resistivity, with total organic carbon (OC) < 40 µg L$^{-1}$). Other reagents include acetonitrile (Optima$^{TM}$, Fisher Scientific), ammonium acetate (≥ 99 %, Fluka, Sigma Aldrich) and ammonium hydroxide (Optima, Fisher Scientific).

### 2. 2 PM$_{2.5}$ sample collection

PM$_{2.5}$ samples were collected in Atlanta, GA from 29 July to 27 August in 2015. A medium volume sampler (3000B, URG Corp.) operated at a flow rate of 90 L min$^{-1}$ was used to collect PM$_{2.5}$ on pre-baked (550 ˚C for 18 hours) quartz fiber filters (90 mm, Pallflex® Tissuquartz™, Pall life science). The PM$_{2.5}$ sampler was placed on the roof top of the School of Earth and Atmospheric Sciences building at the Georgia Institute of Technology (33°46'44.2" N, 84°23'46.2" W; height ~30-40 m). A detailed description of the sampling site is provided by Verma et al. (2014). Samples were collected daily from 1:30 p.m. to 12:30 p.m. next day (local time). One filter blank was collected for every five PM$_{2.5}$ samples. Samples from 29 July, 03, 11, and 19 August were not analysed for organosulfates as the filters were used for a different purpose. The collected samples were placed in aluminium-lined (pre-baked at 550 ˚C for 18 hours) Petri dishes, sealed with Teflon tape, and stored at -20 ˚C until extracted.

### 2.3 Extraction of organosulfates

Organosulfates were extracted according to the method described in Hettiyadura et al. (2015) that has been demonstrated to efficiently recover 83-121% of organosulfates with aliphatic, aromatic, carbonyl, hydroxyl, and carboxyl acid groups. Briefly, sub-samples of filters (averaging ~3 cm$^2$) were extracted with 10.0 mL of acetonitrile and ultra-pure water (95:5, by volume) for 20 minutes by ultra-sonication (5510, Branson). The sample extracts were filtered using polypropylene membrane syringe filter disks (0.45 µm pore size, Puradisc™25 PP, Whatman®). The extracts were





evaporated to dryness under ultra-high purity nitrogen gas at 50 ˚C (Turbovap®LV, Caliper Life Sciences, Reacti-Therm III TS 18824, and Reacti-Vap I 18825, Thermo Scientific). Dried extracts were reconstituted in 600 µL of acetonitrile and ultra-pure water (95:5 by volume).

## 2. 4 Quantification of organosulfates

Organosulfates were quantified using HILIC coupled with negative electrospray ionization tandem mass spectrometry (ESI(-)-MS/MS)  on an ultra-performance liquid chromatography system (UPLC, ACQUITY UPLC H-Class, Waters) coupled with a triple quadrupole (TQ) mass spectrometer (AQCUITY, Waters). The separation of organosulfates was performed on an ethylene-bridged hybrid amide column using an acetonitrile rich mobile phase (acetonitrile and ultra-pure water; 95: 5) and an aqueous mobile phase (ultra-pure water; 100%). Both mobile phases were buffered at pH 9 with 10

mM ammonium acetate and ammonium hydroxide. Organosulfates were eluted using a stepwise gradient as described in Hettiyadura et al. (2015). Targeted analysis was performed in multiple reaction monitoring mode. Hydroxyacetone sulfate and glycolic acid sulfate were quantified using authentic standards. Lactic acid sulfate and methyltetrol sulfate were quantified using their response factors determined previously using authentic standards. Notably, these prior experiments had response factors (determined as the slope of the calibration curve) for glycolic acid sulfate and hydroxyacetone sulfate that

were within 10% of the current experiments, indicating that instrument performance and ionization were consistent within 10%. The optimized ESI(-)-MS/MS conditions used for each of these organosulfates are given in  Hettiyadura et al. (2015) and Hettiyadura et al. (2018), respectively.

Semi-quantitation of other organosulfates was based upon the MS/MS response of authentic standards and matched to the sulfur-containing fragment ions observed. For semi-quantitation of organosulfates that fragmented to the bisulfate

anion ($m/z$ 97, Fig. 1a), $m/z$ 211, 213, and 260  the response factor of 2-methyltetrol sulfate was used, other organosulfates eluting prior to four minutes hydroxyacetone sulfate was used, and those retaining more than four minutes glycolic acid sulfate was used. For the semi-quantitation of organosulfates that fragmented only to the sulfate radical anion ($m/z$ 96, Fig. 1b), methyl sulfate was used. Organosulfates with $m/z$ 137, 139 and 296 that fragmented to the sulfite radical anion ($m/z$ 80, Fig. 1d) hydroxyacetone sulfate was used. The cone voltage and collision energy used for the organosulfates that were semi-

quantified using surrogate standards were same as the ESI(-)-MS conditions used for corresponding precursor ion scans (given in Sect. 2.5.1). The uncertainty of the organosulfate concentrations were calculated accounting for relative errors in air volume, extraction efficiency and instrumental analysis according to the method described in Hettiyadura et al. (2017). The relative error in the instrument analysis was propagated using the limit of detection and the relative standard deviation for each organosulfate standard given in Hettiyadura et al. (2015). For methyltetrol sulfate and the other organosulfates that

did not have standards, 30% of the measurement was used as an estimate of the analytical uncertainty (Hettiyadura et al., 2018). This uncertainty does not account for any bias introduced by the use of a surrogate standard, which can be only evaluated by using an authentic standard. All data were acquired and analyzed using MassLynx and QuanLynx softwares (Waters Inc., version 4.1).



## 2.5 Qualitative analysis of organosulfates

### 2.5.1 Precursor ion scans

Sample analysis was performed on the UPLC-TQ in precursor ion mode as described in Hettiyadura et al. (2017). Briefly, a cone voltage and a collision energy of 28 V and 16 eV were used for the $m/z$ 97 precursor ion scan and 42 V and 20 eV were used for the $m/z$ 96 precursor ion scan. In addition, precursor ion scans of $m/z$ 81 (bisulfite anion) and $m/z$ 80 were used to identify organosulfates that did not fragment into $m/z$ 97 or 96, for which a cone voltage of 34 V and a collision energy of 18 eV were used. A mass range of 100-400 Da was used in all precursor ion scans. The data were acquired and analysed using MassLynx and QuanLynx software packages (Waters Inc., version 4.1).

### 2.5.2 Chemical characterization and structure elucidation

PM extracts were also analyzed by a UPLC-ToF mass spectrometer (Bruker Daltonics MicrOTOF) to determine the elemental composition and structural information of the major sulfur- containing species. The ESI(-) conditions included a capillary voltage of 2.6 kV, a cone voltage of 30 V, and a desolvation gas flow rate of 600 L h$^{-1}$. Other ESI(-) conditions used were the same as in Hettiyadura et al. (2015). Data were collected in a mass range 100-400 Da. A peptide, Val-Tyr-Val ($m/z$ 378.2029, Sigma-Aldrich), was used as the lock mass to correct for any instrument drift. Molecular formulas were assigned considering both odd and even electron states, $C_{1-25}$, $H_{0-50}$, $O_{3-20}$, $S_{1-2}$, $N_{0-5}$, and a maximum error of 10 mDa. The data were acquired and analyzed using MassLynx software (Water, version 4.1) and elemental composition tool (Water Inc., version 4.0).

## 2.6 Collocated measurements

Percent contributions of organosulfates to $PM_{2.5}$ OC were compared to determine the relative abundances of the major organosulfates in Atlanta and Centreville. OC in the $PM_{2.5}$ samples were measured on 1 cm$^2$ filter punches using a thermal-optical analyzer (Sunset laboratory) according to Schauer et al. (2003). Filter-based measurements of other $PM_{2.5}$ components, gas-phase measurements, and meteorological conditions were used in correlation analysis to provide insight to precursors and formation pathways of organosulfates. Isoprene SOA tracers (2-methylthreitol, 2-methylerythritols, 2-methylglyceric acid, cis-2-methyl-1,3,4-trihydroxy-1-butadiene, 3-methyl-2,3,4-trihydroxy-1-butene, and trans-2-methyl-1,3,4-trihydroxy-1-butene), cis-pinonic acid, β-caryophillinic acid, meso-erythritol, 2,3-dihydroxy-4-oxopentanoic acid, aromatic dicarboxylic acids (phthalic acid, terephthalic acid, isophthalic acid, and 4-methylphthalic acid), and mononitroaromatic compounds (4-nitrophenol, 2-methyl-4-nitrophenol, 4-methyl-2-nitrophenol, 4-nitrocatechol, 3-methyl-6-nitrocatechol, and 3-methyl-5-nitrocatechol) were measured by gas chromatography (GC)-MS according to the methods described in Al-Naiema and Stone (2017). Sulfate was measured by ion chromatography following Jayarathne et al. (2014). The hourly-based measurements of $O_3$, $NO_x$ (nitrogen oxides such as, NO and $NO_2$), and solar radiation were obtained from the Southeastern Aerosol and Research Characterization network monitoring site at Jefferson Street (JST) located 2 km west



of the sampling site and were averaged across the sample collection time. Detailed descriptions of their quantification methods are described in Hansen et al. (2003).

**2.7 Correlation analysis**

Pearson's correlation coefficients were assessed using a statistical analysis software (IBM® SPSS® statistics, version 25). Correlations were interpreted as very strong (0.9-1.0), strong (0.7-0.9), moderate (0.5-0.7), weak (0.3-0.5), or negligible (0.0-0.3) (Hinkle et al., 2003). The correlations were considered as statistically significant at the 95% confidence.

**3 Results and Discussion**

**3.1 Quantitative analysis of organosulfates in Atlanta**

Quantitative information about the organosulfates observed in Atlanta are summarized in Table 1, with time series of select species shown in Figure 2. Methyltetrol sulfate is the most abundant quantified organosulfate, contributing 12.6% of $PM_{2.5}$ OC, followed by $m/z$ 211 (0.93%), 213 (0.80%), glycolic acid sulfate (0.24%), 2-methylglyceric acid sulfate (0.32%), and lactic acid sulfate (0.20%) (Table 2). The remaining 26 organosulfates were estimated to contribute 1% of $PM_{2.5}$ OC. Altogether, the 32 measured organosulfates in Table 1 account for 16.5% of $PM_{2.5}$ OC. These results indicate that organosulfates in Atlanta during August 2015 were dominated by methyltetrol sulfate, with minor contributions from many species.

**3.2 Qualitative analysis of major organosulfates in Atlanta**

Organosulfates were identified by precursors to $m/z$ 97 ($HSO_4^-$), 96 ($SO_4^{-}$), 81 ($HSO_3^-$), and 80 ($SO_3^{-}$) in three $PM_{2.5}$ samples collected on 30-31 July and 01 August in 2015. Results were similar for all three samples, therefore the results obtained only for 30 July sample is shown in Figure 1. Major species were defined in one of two ways: 1) having a minimum relative intensity (≥1.0% for $m/z$ 97, >12% for $m/z$ 96, >5% for $m/z$ 81, and >3% for $m/z$ 80 in any of the three samples) or 2) by retaining more than four minutes on the HILIC column that results in a lower relative response due to changing mobile phase conditions, despite high ambient concentrations (Hettiyadura et al., 2017). The absolute MS signal for precursors to $m/z$ 97 was 52, 10, and 8 times greater than $m/z$ 96, 81, and 80, respectively; however, due to differing ionization efficiencies and stabilities among these fragment ions, the strength of the MS signal is not indicative of the relative concentrations of species that form these fragments. Table 1 summarizes the major organosulfates' elemental composition, monoisotopic mass, proposed or known structures and precursor gases. Of the major organosulfates, 26 of the 32 consisted of C, H, O, and S, while 6 of 32 consisted of C, H, O, S, and N. Structures were proposed based on elemental composition, double bond equivalence (DBE), retention time, and prior studies.



### 3.3 Isoprene-derived organosulfates in Atlanta.

The strongest organosulfate signals are attributed to species associated with isoprene SOA. Methyltetrol sulfate (*m/z* 215), the most abundant organosulfate observed, is produced from the acid catalyzed nucleophilic addition of sulfate to IEPOX ring (Surratt et al., 2010). Organosulfates with *m/z* 211 (sulfate esters of methyldihydroxylactone) and 213 (sulfate esters of cyclic methyltrihydroxyaldehyde hemiacetal), of the next-highest abundance, have been observed during photo-oxidation of isoprene (Surratt et al., 2008) and are suggested to derive from oxidation of primary alcohols in methyltetrol sulfates (Hettiyadura et al., 2015). In addition, 14 other major organosulfates identified are known to derive from isoprene and isoprene oxidation products (Table 1). Many of these organosulfates have been identified as SOA products from diesel and biodiesel fuel emissions (e.g., 2-methylglyceric acid sulfate, lactic acid sulfate, hydroxyacetone sulfate, *m/z* 167, 183, 197, 211, 213, 237, 239, and 253) (Blair et al., 2017), monoterpenes (*m/z* 239 and 253) (Surratt et al., 2008), and/or MBO (199; $C_5H_{11}SO_6^-$) (Zhang et al., 2012). However, their moderate to strong correlations with methyltetrol sulfate (Table S1) and 2-methyltetrols (Table S2) suggest that they are mainly derived from isoprene. 2-Methylglyceric acid sulfate correlated significantly with 2-methylglyceric acid (r=0.608, p-value=0.001, Table S2); both of these compounds are tracers for isoprene high-NO$_x$ SOA, and are formed by the acid-catalyzed nucleophilic addition of sulfate and water, respectively, to methacrylic acid epoxide (MAE) or hydroxymethyl-methyl-α-lactone (HMML) (Lin et al., 2013). The organosulfate with *m/z* 260 is a nitooxy-organosulfate that derives from photooxidation of isoprene under high-NO$_x$ conditions (Surratt et al., 2008; Gómez-González et al., 2008). Two isomers of *m/z* 260 were identified in this study, while up to four isomers of *m/z* 260 were reported in Centreville (Surratt et al., 2008). The *m/z* 260 also correlated moderately with methyltetrol sulfate (r=0.539, p-value=0.005, Table S1), supporting its formation from isoprene. The organosulfate with *m/z* 274 is also a nitrooxy organosulfate that is derived from isoprene photooxidation under high-NO$_x$ conditions (Nestorowicz et al., in review). The organosulfate with *m/z* 274 has multiple isomers, while only the two isomers retaining greater than 4 min are considered to be major ones (Sect. 3.2 and Fig. 3o). Their longer retention times (5.6 and 5.8 min), three additional oxygen atoms, and one unit of unsaturation suggest the presence of a carboxylic acid functional group and a hydroxyl group. Plausible structures for these two organosulfates are diastereomers of 2-carboxy-3-hydroxy-4-(nitrooxy)butan-2-yl sulfate (Table 1), which could form by the oxidation of a primary hydroxyl group in 1,3-dihydroxy-2-methyl-4-(nitrooxy)butan-2-yl sulfate (an isomer of *m/z* 260, $C_5H_{10}SO_9^-$, proposed by Darer et al. (2011)) to a carboxylic acid. The strong correlation of these two signals at *m/z* 274 with the less-oxidized isoprene nitrooxy-organosulfate (*m/z* 260) (r=0.860, p-value<0.001, Table S1), supports this prediction. Overall, these results indicate that isoprene is the major precursor of the most abundant organosulfates in this study.

Isoprene-derived organosulfates explain a significant fraction of isoprene-derived organic aerosol observed in Atlanta that had not previously been identified on a molecular level. IEPOX-derived OA accounted for 29% (3.3 µg m$^{-3}$) of PM$_1$ OA at the nearby JST monitoring site in summer 2014, while the measured IEPOX-OA tracers (2-methyltetrols, C$_5$-alkene triols, and 3-methyl-hydrofuran-3,4-diols) in PM$_{2.5}$ (averaged 391.7 ng m$^{-3}$) accounted for 3% of PM$_1$ OA



(Rattanavaraha et al., 2017), assuming negligible differences between $PM_1$ and $PM_{2.5}$. The remaining IEPOX-derived OA corresponded to 10-18% of $PM_1$ OC (considering an OM:OC ratio of $2.05 \pm 0.57$) (Xu et al., 2017), and is comparable to the contribution of isoprene-derived organosulfates to $PM_{2.5}$ OC in this study (15.7%). Additionally, the isoprene-derived organosulfates observed in this study account for more than half of the $PM_{2.5}$ secondary organic carbon coming from

5 isoprene, which is estimated as 27% following the SOA tracer method (Al-Naiema et al., in preparation; Kleindienst et al., 2007). These results indicate that more than half of the isoprene-derived OA in Atlanta during summer is comprised by organosulfates, mainly methyltetrol sulfate.

### 3.4 Monoterpene-derived organosulfates in Atlanta.

    Seven of the 32 major organosulfates identified in Atlanta (Table 1) were previously detected among the SOA

produced from monoterpenes in the presence of $NO_x$ and acidic sulfate seed aerosols (Surratt et al., 2008). Of these, nitroxy-organosulfates at $m/z$ 342, 294, and 296 are derived from monoterpenes either by photooxidation in the presence of $NO_x$ or from nitrate radical-initiated oxidation (Surratt et al., 2008; Iinuma et al., 2007). The estimated contribution of these seven monoterpene-derived organosulfates is 0.5% of $PM_{2.5}$ OC. However, the accuracy of this estimate is limited by the lack of authentic standards for monoterpene organosulfates and the large differences in molecular structure between the

monoterpene organosulfates and the standards utilized in this study. The absence of significant correlations among nitroxy-organosulfates with other organosulfates (Table S1) and biogenic SOA tracers that predominantly derive from photooxidation reactions (Table S2) suggest that these nitroxy-organosulfates likely formed by nitrate radical-initiated oxidation. Organosulfates with $m/z$ 223, 279, and 281 have been identified as SOA products of α-pinene, as well as from other monoterpenes ($m/z$ 279 and 281), in the presence of $NO_x$ and highly acidic sulfate seed aerosol (Surratt et al., 2008).

The organosulfate with $m/z$ 251 has been identified in SOA from the photooxidation of β-caryophyllene (a sesquiterpene) and limonene (a monoterpene) in the presence of $NO_x$ and sulfate seed aerosols (Chan et al., 2011; Surratt et al., 2008). These species did not correlate with β-caryophyllinic acid (Table S2), an SOA tracer for β-caryophyllene formed under high-$NO_x$ conditions (Jaoui et al., 2007), suggesting that $m/z$ 251 mainly forms from monoterpenes. Organosulfates with the same $m/z$ were also detected among the organosulfates generated from diesel and biodiesel fuel emissions (Blair et al., 2017) and

photooxidation of $n$-alkanes such as decaline ($m/z$ 281) and cyclodecane ($m/z$ 279 and 281) (Riva et al., 2016a), but these species are expected to be biogenic in nature due to dominance of biogenic VOC in Atlanta during summer (Geron et al., 1995; Al-Naiema et al., in preparation; Rattanavaraha et al., 2017).

### 3.5 Organosulfates derived from diesel and/or biodiesel fuel emissions in Atlanta

    Five organosulfates that were previously reported only in photooxidation of diesel and/or biodiesel fuel in the

30 presence of $SO_2$ were identified among the 32 major organosulfates. These include $m/z$ 137 and 151 that were generated from diesel fuel emissions and $m/z$ 195, 209, and 265 that were generated from both diesel and biodiesel emissions (Blair et al., 2017). The organosulfate with $m/z$ 265 corresponds to dodecyl sulfate, a widely used surfactant in detergents that can

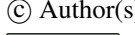


also come from wastewater treatment plants (Hettiyadura et al., 2017). The concentrations of $m/z$ 209 and 195 are at least 3 times higher compared to other organosulfates derived from diesel and/or biodiesel emissions in this study (Table 1). These organosulfates ($m/z$ 209 and 195) were also detected with a high abundance in urban Shanghai and Los Angeles (Tao et al., 2014). The organosulfates with $m/z$ 209 and 195 are homologs, differing by one methylene. Both compounds have two units

of unsaturation and two additional oxygen atoms. Further, their retention times, which were less than a minute, suggest that they do not contain a carboxylic acid group, but may contain two carbonyl groups (Fig. S3). Additional work is required to determine the position of carbonyl and sulfate groups in these compounds. As $m/z$ 209 and 195 are highly abundant in other urban locations and are only known to derive from diesel and/or biodiesel fuel, they may be useful as tracers for SOA derived from diesel and biodiesel emissions.

**3.6 Aromatic organosulfates in Atlanta**

Aromatic sulfur-containing compounds were not detected among the major organosulfate species (Table 1), although some were observed by ToF MS. Two sulfur-containing compounds had large DBEs indicating aromatic groups: $m/z$ 185 ($t_R$ 1.06 min, $C_7H_5SO_4^-$, DBE 5.5, error 3.7 mDa) and 201 ($t_R$ 7.56 and 8.17 min, $C_7H_5SO_5^-$, DBE 5.5, error 3.5 mDa). The MS data matched the molecular formula reported by Riva et al. (2015), who detected $m/z$ 185 in naphthalene and

2-methylnaphthalene photooxidation experiments and identified it as formylbenzenesulfonate by MS fragmentation. Riva et al. (2015) also reported $m/z$ 201 in SOA generated by the photooxidation of 2-methylnaphathelene and identified it as 4-sulfobenzoic acid using an authentic standard. In the Atlanta PM$_{2.5}$, two isomers of $m/z$ 201, likely conformational isomers of 4-sulfobenzoic acid, are observed. The presence of a carboxylic acid group in $m/z$ 201 is evident by the retention time > 7 minutes in the HILIC method (Hettiyadura et al., 2015). None of the aromatic organosulfates reported in Staudt et al. (2014)

(phenyl sulfates and benzyl sulfates) were detected in ToF MS. This may be due to the lower retention times and higher detection limits for aromatic organosulfates in HILIC compared to reversed-phase LC (Hettiyadura et al., 2015). These results suggest that aromatic organosulfates have low PM$_{2.5}$ concentrations in comparison to biogenic organosulfates in Atlanta during the summertime.

**3.7 New organosulfates identified in Atlanta**

Three major organosulfates observed at $m/z$ 155 ($C_3H_7SO_4^-$), 165, and 242 have not been previously reported in laboratory smog chamber experiments. They were detected in the PM$_{2.5}$ collected from Centreville during summer 2013 (Hettiyadura et al., 2017). Of these, the $m/z$ 155 was identified as a mono-hydroxy propyl sulfate and was among the ten major organosulfate signals identified in Centreville (Hettiyadura et al., 2017). The $m/z$ 155 in Atlanta correlated with most of the isoprene-derived organosulfates (Table S1), suggesting that it was derived from isoprene. Insight into the chemical

composition, structure, and origins of $m/z$ 165 and 242 are presented in the following paragraphs.

The organosulfate at $m/z$ 165 has an elemental composition of $C_4H_5SO_5^-$, indicating the presence of sulfate, an additional oxygenated functional group, and two DBEs. The ToF chromatograms (Fig. 3d) indicate two isomers of $m/z$ 165



that eluted < 2 min. While both isomers were fragmented into $m/z$ 80, only the first isomer fragmented into $m/z$ 96, which was quantified. Its elemental composition and DBE suggest a dihydrofuran ring structure (Table 1). The strong correlations of $m/z$ 165 with methyltetrol sulfate (r=0.720, p-value<0.001; Table S1) and 2-methyltetrols (r=0.670 and 0.768, p-value<0.001; Table S2) suggest that it also derived from isoprene.

5          The organosulfate at $m/z$ 242 has an elemental composition of $C_5H_8NSO_8^-$, indicating the presence of sulfate, nitrooxy, an oxygenated functional group, and two DBEs. Its short retention time of 0.5 min (Fig. 3k) suggests that it contains a carbonyl group as organosulfates with hydroxyl and carboxylate groups retain more than 1 and 4 minutes, respectively (Hettiyadura et al., 2015; Hettiyadura et al., 2017). A possible formation pathway for this nitrooxy-organosulfate can be loss of a water molecule from 2,3-dihydroxy-3-methyl-4-(nitrooxy)butyl sulfate (an isomer of $m/z$ 260, 10   $C_5H_{10}SO_9^-$, proposed by Gómez-González et al. (2008)) forming an enol that tautomerizes to its stable carbonyl form resulting in 3-methyl-4-(nitrooxy)-2-oxobutyl sulfate (Table 1). Only a few atmospherically relevant isoprene-derived nitrooxy organosulfates have been identified in previous studies. These include $m/z$ 244, 260, 274, and 305 that are derived from isoprene photooxidation under high-$NO_x$ conditions (Surratt et al., 2008; Gómez-González et al., 2012). It is expected that $m/z$ 242 is an additional nitrooxy organosulfate that has not been previously identified in isoprene photooxidation 15   experiments. As this nitrooxy organosulfate is expected to derive from $m/z$ 260, it may useful as a tracer for isoprene SOA formed under high-$NO_x$ conditions and provide insights into atmospheric aging of isoprene-derived SOA.

### 3.8 Comparison of major organosulfates in Atlanta and Centreville

         To better understand the extent of which anthropogenic pollutants influence biogenic SOA formation in urban Atlanta during summer, the concentrations of the major organosulfates were compared to those measured in rural 20   Centreville, AL during summer 2013 analyzed by similar methodology (Hettiyadura et al., 2017; Hidy et al., 2014). Although the major organosulfates identified in both sites were similar and mainly derived from isoprene, their concentrations were two to six times higher in Atlanta than in Centreville, with greatest enhancement obtained for 2-methylglyceric acid sulfate (Table 2). Since the absolute concentrations of these organosulfates can vary with time due to changes in meteorology and other factors influencing their formation, their relative contributions to $PM_{2.5}$ OC were 25   compared across the two sites (Table 2). In total, 12 organosulfates quantified or semi-quantified in both studies contributed 7% of $PM_{2.5}$ OC in Centreville, and 16% in Atlanta. These 12 organosulfates accounted for 95% of the total organosulfate mass in Atlanta and 58-78% of the total bisulfate ion signal in Centreville (Hettiyadura et al., 2017), indicating that these were the dominant species at both sites. Similarly, the IEPOX-OA in Atlanta during August 2012 (31% of $PM_1$ OA) was ~two times greater than IEPOX-OA in Centreville in summer 2013 (18% of $PM_1$ OA) (Xu et al., 2015a; Xu et al., 2015b). 30   Overall, these results suggest isoprene SOA is two times higher in Atlanta compared to Centreville during summer.

         Correlations of major organosulfate species were examined at both Atlanta and Centreville sites to gain insight to their sources and formation pathways. Organosulfates at both sites show moderate to strong correlations with isoprene, isoprene oxidation products, and/or isoprene SOA tracers (Table S2; Table S6 in Hettiyadura et al. (2018)), supporting that



they mainly derive from isoprene. The correlations of sulfate with most of the organosulfates were weak or negligible in Atlanta (Table S4), but were moderate to strong in Centreville (r = 0.5-0.8) (Table S6 in Hettiyadura et al. (2018)). This is likely due to the consistently high levels of sulfate observed in urban Atlanta (ranging $0.82 – 3.24$ µg m$^{-3}$, averaging $1.70 \pm 0.58$ µg m$^{-3}$) compared to more variable sulfate concentrations in rural Centreville (ranging $0.42 – 4.17$ µg m$^{-3}$, averaging $1.78 \pm 0.81$ µg m$^{-3}$) (Hettiyadura et al., 2017). Overall these results suggest isoprene and sulfate are important factors influencing the organosulfate formation in both urban Atlanta and rural Centreville.

Isoprene-derived organosulfates indicated a stronger influence of $NO_x$ on their formation in Atlanta compared to Centreville. A $NO_x$ influence is evident by the elevated levels of high-$NO_x$ isoprene oxidation products such as 2-methylglyceric acid sulfate, which was six times higher in Atlanta than in Centreville, and the isoprene-derived nitrooxy organosulfate at $m/z$ 260 being the 8$^{th}$ strongest organosulfate signal in Atlanta. These results are consistent with the average $NO_x$ concentration in urban Atlanta in August 2015 (10.5 ppb) that was 15 times greater than the average $NO_x$ concentration in rural Centreville during summer 2013 (0.7 ppb) (SOAS, 2013). Methyltetrol sulfate, the most abundant organosulfate at both sites, is thus expected to derive from low-$NO_x$ oxidation pathway in Centreville as described in Surratt et al. (2010) and by high-$NO_x$ oxidation pathway in Atlanta as described in Jacobs et al. (2014). The moderate and strong correlations obtained for isoprene-derived organosulfates with high-$NO_x$ SOA products (Table S3) such as meso-erythritol (Angove et al., 2006) and nitroaromatic compounds (Al-Naiema and Stone, 2017), as well as with ozone (Table S4) that is formed by the photochemical reactions of $NO_x$ and VOC (Blanchard et al., 2014), support that $NO_x$ play a key role in isoprene-derived organosulfate formation in Atlanta. Together, these results suggest a greater influence of $NO_x$ on isoprene-SOA formation in Atlanta compared to Centreville in summer.

## 4 Implications and future work

This study provides insights to the major organosulfate species that should be targets for future measurements and standard development. The three most abundant organosulfates measured in both Atlanta and Centreville include methyltetrol sulfate, $m/z$ 211, and 213. Of these, a standard for methyltetrol sulfate was synthesized (Budisulistiorini et al., 2015; Bondy et al., 2018). Six isomers of methyltetrol sulfates were baseline resolved in the PM$_{2.5}$ samples collected from Centreville and Atlanta. Based on their stability to acid hydrolysis, these were tentatively identified as diastereomer pairs of methyltetrol sulfates with the sulfate group attached to primary (highest stability), secondary (intermediate stability), and tertiary (lowest stability) carbons (Hettiyadura et al., 2017). Development of standards for quantification of the three methyltetrol diastereomer pairs thus will give insights to atmospheric aging and lifetime of this compound. Given the ubiquity and high abundance of $m/z$ 211 and 213 in the Southeastern US and other locations (Hettiyadura et al., 2017; Spolnik et al., 2018), they should be the next highest priorities for standard development. The $m/z$ 211 and 213 also have multiple isomers as described by Hettiyadura et al. (2015) and Spolnik et al. (2018). Further, this study reveals isoprene-




derived organosulfates such as 2-methylglyceric acid sulfate and *m/z* 260 are useful in distinguishing SOA formed under high-NO$_x$ conditions in urban environments.

While isoprene was the major precursor to organosulfates at both Atlanta and Centreville, the comparison of these two datasets reveals different anthropogenic influences on biogenic SOA formation (Sect. 3.8). In particular, NO$_x$ had a stronger influence on organosulfate formation in Atlanta and sulfate having a stronger influence on organosulfate formation in Centreville. Future studies should focus on comparing the major organosulfate species in other urban and rural locations in the Southeastern US to determine if these trends are ubiquitous across urban-rural landscapes and to better understand the anthropogenic influences on biogenic SOA formation. While high levels of isoprene-derived organosulfates detected in the Southeastern US during summer coincide with high isoprene emissions from plants, high levels of aromatic organosulfates and nitrooxy organosulfates detected in fall and winter coincide with high levels of biomass burning (Ma et al., 2014; He et al., 2014). Thus, longer-term measurements of organosulfates spanning an annual cycle are needed to further evaluate the sources and concentrations of organosulfates in the atmosphere.

## 5 Data availability

Organosulfate measurements are given in Table S5 and other PM$_{2.5}$ measurements used in this study will be published soon elsewhere (Al-Naiema et al., *in preparation*).

## 6 Disclaimer

Any opinions, findings, and conclusions or recommendations expressed in this material are those of the author(s) and do not necessarily reflect the views of the National Science Foundation (NSF).

## 7 Acknowledgements

The authors would like to thank E. Geddes, K. Richards, and T. Humphrey at the Truman State University for synthesizing standards of hydroxyacetone sulfate and glycolic acid sulfate; S. Staudt at the University of Wisconsin, Madison for synthesizing the lactic acid sulfate standard; J. D. Surratt, A. Gold and Z. Zhang at the University of North Carolina at Chapel Hill for providing 2-methyltetrol sulfate standard; J. Kettler and C. Madler for their assistance in sample preparation and analysis; L. Teesch and V. Parcell for their assistance in the University of Iowa High Resolution Mass Spectrometry Facility (HRMSF); and R. J. Weber for assistance with sample collection. This research was supported by the National Science Foundation AGS grant number 1405014.



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





**Figure Captions**

**Figure 1.** Precursors of a) bisulfate ion (*m/z* 97), b) sulfate ion radical (*m/z* 96), c) bisulfite ion (*m/z* 81), and d) sulfite ion radical (*m/z* 80) identified from a sample collected on 30 July 2015 in Atlanta. Blue indicates nominal *m/z* of the major organosulfate species that were identified from the precursor *m/z* 97 scan. Red indicates nominal *m/z* of the major organosulfate species that were identified from the precursor *m/z* 96, 81 and 80 scans.

**Figure 2.** Time series of seven major organosulfate species quantified in August 2015 in Atlanta. Samples that were not analyzed (because they were used for a different purpose) are marked with a star.

**Figure 3.** Extracted chromatograms of 19 major organosulfate species obtained from a PM$_{2.5}$ sample collected in Atlanta using HR-ToF (at 0.01 Da error). Extracted chromatograms of the remaining 13 major organosulfate species are shown in Hettiyadura et al. (2017) for a PM$_{2.5}$ sample collected in Centreville. MS data, structures, and VOC precursors of these organosulfates are given in Table 1.



**Table 1**. The major organosulfates identified using HILIC-TQ in daily PM$_{2.5}$ samples collected from Atlanta, GA in August 2015, indicating nominal mass-to-charge ratio (*m/z*), chemical formula and monoisotopic mass (at 0.01 Da) determined from HILIC-ToF, proposed structure (with a star indicating many isomers, although only one is shown), and potential VOC precursors, and their average ambient concentrations with one standard deviation (SD). For these organosulfates the median and the maximum error in the observed mass is 1.7 and 7.5 mDa, respectively. Organosulfates are ordered in the table from greatest to least abundance.

| *m/z* [M-H]⁻ | Formula [M-H]⁻ | Monoisotopic mass [M-H]⁻ | Proposed structure | Precursor(s) | Average (SD) (ng m⁻³) |
|---|---|---|---|---|---|
| 215 | C$_5$H$_{11}$SO$_7^-$ | 215.0225 (Methyltetrol sulfate) | [1]* <br>  | Isoprene[2-5] | 1791.7 (1084.7) |
| 211 | C$_5$H$_7$SO$_7^-$ | 210.9912 | [6]* <br>  | Isoprene[2] | 130.6 (81.9)[7] |
| 213 | C$_5$H$_9$SO$_7^-$ | 213.0069 | [6]* <br>  | Isoprene[2-3] | 114.3 (78.9)[7] |
| 155 | C$_2$H$_3$SO$_6^-$ | 154.9650 (Glycolic acid sulfate) | [6-8] <br>  | Isoprene,[2, 4-5] MVK[4, 9] | 58.5 (40.2) |
| 199 | C$_4$H$_7$SO$_7^-$ | 198.9912 | [2] <br>  | Isoprene,[2-5] MVK and MACR[4, 9] | 53.0 (42.3)[10] |



| 169 | $C_3H_5SO_6^-$ | 168.9807 (Lactic acid sulfate) [6-8] | Isoprene,[2-3, 5] 3-E-hexenal, 3-Z-hezenal, and ]2-E-pentenal,[11] MVK[4] | 38.4 (24.2) |
|---|---|---|---|---|
| 183 | $C_4H_7SO_6^-$ | 182.9963 [12]* | Isoprene,[3] MACR and MVK[9] | 23.4 (14.9)[13] |
| 260 | $C_5H_{10}NSO_9^-$ | 260.0076 [14]* | Isoprene[2, 5] | 18.7 (11.2)[7] |
| 197 | $C_5H_9SO_6^-$ | 197.0120 [3]* | Isoprene[3] | 13.3 (6.1)[13] |
| 281 | $C_{10}H_{17}SO_7^-$ | 281.0695 [15]* | Monoterpenes,[2] pinene[4] | 12.1 (7.8)[13] |
| 239 | $C_7H_{11}SO_7^-$ | 239.0225 [4]* | Limonene,[2] MVK and MACR[4] | 11.5 (6.1)[13] |
| 209 | $C_6H_9SO_6^-$ | 209.0120 *Not identified DBE 2.5 | Diesel and biodiesel fuel[16] | 10.2 (6.3)[13] |





| 153 | $C_3H_5SO_5^-$ | 152.9858 (Hydroxyacetone sulfate) | [2] 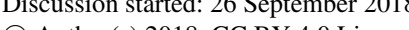 | Isoprene,[2-3] MACR and MVK[9] | 10.1 (6.0) |
| 294 | $C_{10}H_{16}NSO_7^-$ | 294.0647 | [2]* | α-Pinene, terpinolene, and α-terpinene,[2] β-pinene[2, 17] | 9.0 (7.1)[18] |
| 199 | $C_5H_{11}SO_6^-$ | 199.0276 | [19]* | Isoprene,[3] MBO[19] | 8.4 (5.4)[13] |
| 251 | $C_9H_{15}SO_6^-$ | 251.0589 | [2]* | Limonene,[2] β-caryophilline[20] | 8.0 (3.3)[13] |
| 195 | $C_5H_7SO_6^-$ | 194.9963 | *Not identified DBE 2.5 | Diesel and biodiesel fuel[16] | 7.6 (4.5)[13] |
| 342 | $C_{10}H_{16}NSO_{10}^-$ | 342.0495 | [21] | α-Pinene and α-terpinene,[2] β-pinene,[2, 17] | 7.1 (3.9)[13] |




| 279 | $C_{10}H_{15}SO_7^-$ | 279.0538 | [2]* | Monoterpenes,[2] pinene[4] | 7.1 (3.2)[13] |
|---|---|---|---|---|---|
| 237 | $C_7H_9SO_7^-$ | 237.0069 | [4]* | MVK[4] | 6.6 (3.2)[13] |
| 223 | $C_7H_{11}SO_6^-$ | 223.0276 | [21]* | α-Pinene[2] | 6.3 (2.8)[13] |
| 253 | $C_8H_{13}SO_7^-$ | 253.0382 | [9]* | α-Terpinene,[2] MVK and MACR[4, 9] | 6.3 (2.7)[13] |
| 167 | $C_4H_7SO_5^-$ | 167.0014 | [9]* | MACR and MVK[9] | 4.2 (1.8)[13] |
| 274 | $C_5H_8NSO_{10}^-$ | 273.9869 | * | Isoprene[22] | 2.9 (1.2)[10] |
| 151 | $C_4H_7SO_4^-$ | 151.0065 | Not identified DBE 1.5 | Diesel[16] | 2.7 (1.9)[18] |





| | | | | | |
|---|---|---|---|---|---|
| 139 | $C_2H_3SO_5^-$ | 138.9701 | [2]* | Isoprene[2] | 2.4 (1.1)[23] |
| 265 | $C_{12}H_{25}SO_4^-$ | 265.1474 | [24] | Diesel and biodiesel fuel[16] | 2.3 (1.5)[13] |
| 165 | $C_4H_5SO_5^-$ | 164.9858 | * | Unknown | 2.0 (1.4)[18] |
| 137 | $C_3H_5SO_4^-$ | 136.9909 | *Not identified DBE 1.5 | Diesel[16] | 1.8 (0.8)[23] |
| 155 | $C_3H_7SO_5^-$ | 155.0014 | [24] | Unknown[24] | 1.6 (0.9)[13] |
| 242 | $C_5H_8NSO_8^-$ | 241.9971 | | Unknown | 0.5 (0.4)[18] |
| 296 | $C_9H_{14}NSO_8^-$ | 296.0440 | [2] | Limonene[2] | 0.5 (0.2)[23] |

Methylvinyl ketone (MVK), methacrolein (MACR), 2-methyl-3-buten-2-ol (MBO), double bond equivalence (DBE); [1]Surratt et al. (2010), [2]Surratt et al. (2008), [3]Riva et al. (2016b), [4]Nozière et al. (2010), [5]Gómez-González et al. (2008), [6]Hettiyadura et al. (2015), [7]quantified using a response factor of $m/z$ 97 of the 2-methyltetrol sulfate standard detected in a previous experiment, [8]Olson et al. (2011), [9]Schindelka et al. (2013), [10]quantified against $m/z$ 97 of glycolic acid sulfate standard, [11]Shalamzari et al. (2016), [12]Shalamzari et al. (2013), [13]quantified against $m/z$ 97 of hydroxyacetone sulfate standard, [14]Darer et al. (2011), [15]Riva et al. (2016a), [16]Blair et al. (2017), [17]Iinuma et al. (2007), [18]quantified against $m/z$ 96 of methyl sulfate standard, [19]Zhang et al. (2012), [20]Chan et al. (2011), [21]Yassine et al. (2012), [22]Nestorowicz et al. (in review), [23]quantified against $m/z$ 80 of hydroxyacetone sulfate standard, [24]Hettiyadura et al. (2017).





**Table 2**. Comparison of organosulfates quantified or semi-quantified in Centreville, AL from 13 June to 13 July, 2013 and in Atlanta, GA in August 2015. Standard deviations are given in parenthesis.

| Organosulfate | Atlanta, GA | | Centreville, AL[1] | |
|---|---|---|---|---|
| | Average (ng m$^{-3}$) | %OC | Average (ng m$^{-3}$) | %OC |
| Hydroxyacetone sulfate ($m/z$ 153)[2] | 10.1 (6.0) | 0.06 (0.03) | 5.8 (3.1) | 0.05 (0.04) |
| Glycolic acid sulfate ($m/z$ 155)[2] | 58.5 (40.2) | 0.24 (0.14) | 20.6 (14.3) | 0.10 (0.08) |
| $C_3H_7SO_5^-$ ($m/z$ 155)[3] | 1.6 (0.9) | 0.01 (0.01) | 1.1 (0.8) | 0.01 (0.01) |
| Lactic acid sulfate ($m/z$ 169)[2] | 38.4 (24.2) | 0.20 (0.11) | 16.5 (10.3) | 0.12 (0.10) |
| $C_4H_7SO_6^-$ ($m/z$ 183)[3] | 23.4 (14.9) | 0.15 (0.07) | 9.4 (5.8) | 0.09 (0.08) |
| $C_4H_7SO_7^-$ ($m/z$ 199)[4] | 53.0 (42.3) | 0.32 (0.22) | 8.4 (9.0) | 0.07 (0.09) |
| $C_5H_{11}SO_6^-$ ($m/z$ 199)[3] | 8.4 (5.4) | 0.06 (0.03) | 2.6 (2.2) | 0.03 (0.03) |
| $C_5H_7SO_7^-$ ($m/z$ 211)[5] | 130.6 (81.9) | 0.93 (0.48) | 35.3 (25.6) | 0.33 (0.31) |
| $C_5H_9SO_7^-$ ($m/z$ 213)[5] | 114.3 (78.9) | 0.80 (0.48) | 31.6 (22.5) | 0.30 (0.26) |
| Methyltetrol sulfate ($m/z$ 215)[2] | 1791.7 (1084.7) | 12.55 (6.25) | 668.2 (515.4) | 6.06 (5.49) |
| $C_7H_{11}SO_7^-$ ($m/z$ 239)[3] | 11.5 (6.1) | 0.10 (0.04) | 7.0 (3.9) | 0.09 (0.07) |
| $C_{10}H_{16}NSO_{10}^-$ ($m/z$ 342)[3] | 7.1 (3.9) | 0.07 (0.04) | 5.7 (5.7) | 0.08 (0.10) |
| *Sum* | 2248.6 | 15.5 | 812.1 | 7.3 |

[1]Published in Hettiyadura et al. (2018), [2]quantified against authentic standards or response factors detected in a previous experiment, [3]semi-quantified against hydroxyacetone sulfate, [4]semi-quantified against glycolic acid sulfate, [5]semi-quantified against 2-methyltetrol sulfate or using its response factor.



Figure 1



Figure 2





Figure 3



**Retention time (min)**