# Peer review of "Organosulfates in Atlanta, Georgia: Anthropogenic influences on biogenic secondary organic aerosol formation"

_Atmospheric Chemistry and Physics, 2018_

## Referee Comment (RC1) · Anonymous Referee #2 · 26 Oct 2018

General comments

The study examines the influence of anthropogenic emissions to formation of organosulfates in an urban site in the Southeastern United States (SE-US). This is achieved by quantifying the major organosulfates from ambient measurements and comparing the identified compounds with those from laboratory experiment and a rural site in the SE-US, and other tracers measurements. The study reveals an enhanced formation of isoprene-derived organosulfate concentration, particularly the 2-methylglyceric acid sulfate which is known to be a tracer for high-NOx isoprene-derived secondary organic aerosol (SOA) formation mechanism. The general objective is clear

and the methods are well executed. There are some minor errors in the manuscript and some clarifications needed. Overall, I recommend accepting this manuscript for publication after corrections as detailed in the following.

Technical comments

Define abbreviations at their first appearance when you are using them repeatedly, such as: GA (Pg 1 Ln 12), AL (Pg 1 Ln 22)

Pg 1 Ln 11: add (US) after Southeastern United States

Pg 7 Lns 12-15: It is said here that the correlation between 2-methylglyceric acid sulfate with 2-methylglyceric acid is significant at r = 0.608. Based on description in Section 2.7, this correlation values is classified to be moderate. The use of "significant" is ambiguous, as it may be inferred as "strong". I recommend being consistent with the classification and description throughout the text.

Pg 11 Ln 30: insert "an authentic" before standard development.

---

## Referee Comment (RC2) · Anonymous Referee #1 · 18 Nov 2018

The manuscript presents new measurements focusing on occurrence of organosulfates in aerosol samples collected in the city of Atlanta, USA. A detailed chemical analysis is reported, while the study would benefit from a more thorough discussion of relation to atmospheric parameters and broader implications for other regions. Such a discussion should cover influence of air mass transport from precursor source areas.

Specific comments

Abstract (page 1 line 26). I suggest to briefly explain the formation mechanisms of organosulfate, as this knowledge is needed to understand your point here. The abstract would also benefit from an introduction in the first sentences to put the work into context

for the general reader.

Introduction.

The introduction needs a more thorough presentation of state of the art and previous studies of organosulfates in aerosols.

Page 2 Line 2: Suggest to rephrase to "Organosulfates are components of atmospheric secondary organic aerosol...".

P2 L3: "Organosulfates have been detected all around the world". Please be more specific and include references.

P2 L6: VOC are in the gasphase while sulfate radicals are in the particle phase.

P2 L8: "largely biogenic" - there have been many studies of biogenic precursors, and fewer of anthropogenic so this statement may not be correct because we have not yet investigated all precursors of organosulfates.

P2 L14: The influence of anthropogenic VOC has been established in e.g. Asian studies.

Experimental

P4 L30: What do you mean by "30% of the measurement"? Of the value?

Results and Discussion

All headlines include the words "in Atlanta". I suggest to leave this out and only specify the comparison with Centreville in that specific headline.

The time series in Figure 2 would benefit from showing time series of relevant pollution indicators.

P6 L14: "many species" - can you be more specific e.g. "many other organosulfate species originating from xxx"

P6 L19-20: Please describe more clearly what you mean by these relative intensities (in the mass spectra, I assume?) of organosulfate and sulphonate species.

P7 L21: As all readers might not be familiar with HILIC, I suggest to explain this reasoning in more details.

P7 The first paragraph contains a lot of information and I suggest to divide into at least two paragraphs to improve readability.

P7 L2: Which "signal" do you refer to?

P7 L4-5: Please check that all compounds names are written correctly - it seems that spaces are missing in some names.

P7 L8: suggest to rephrase to "have also been identified as.."

P7 L16: nitooxy -> nitrooxy

P7 L30. Please explain how "isoprene-derived organic aerosol" was identified and quantified. Using AMS data?

P8 L32: The dodecyl sulfate from surfactants is not covered well by the title of the section.

P9 L24: It is not so clear which organosulfates are detected for the first time ever and which are detected for the first time in Atlanta.

P10 L15-16: This would need additional laboratory studies to confirm.

P10 Section 3.8: Please make it more clear that you are comparing samples collected in different years. Differences in levels are thus heavily affected by differences in meteorology affecting isoprene emissions, transport and mixing of biogenic and anthropogenic pollutants.

P11 L1-5: I suggest to specify that you measure inorganic sulfate as opposed to total sulfate measured by AMS.

P11 L17: Can other species or photochemical processes related to NOx be ruled out?

P11 L19: I suggest to compare your results with some of the previous results obtained during SOAS.

P11 L22: Do you mean standard synthesis (instead of development)?

P11 L23: Suggestion: "Of these, a standard for methyltetrol sulfate was synthesized" -> "Of these, only a standard for methyltetrol sulfate was previously synthesized".

P12 L15: I suggest to divide this into two sentences to indicate more clearly which data are in the SI and which are being published elsewhere.

Table 1: Please check that the resolution of molecular structures is adequate.

Table 2: The number of significant figures should indicate the uncertainty for each number and does not have to be the same for all levels of concentrations. The number of significant figures should thus be reduced for the highest concentrations.

[Figure]

---

## Author Comment (AC1) · 2 Jan 2019

Anonymous Referee # 1, General Comments: "The manuscript presents new measurements focusing on occurrence of organosulfates in aerosol samples collected in the city of Atlanta, USA. A detailed chemical analysis is reported, while the study would benefit from a more thorough discussion of relation to atmospheric parameters and broader implications for other regions. Such a discussion should cover influence of air mass transport from precursor source areas."

Authors' response to anonymous referee # 1, general comments: We thank the referee for his/her thoughtful comments and suggestions to improve the manuscript. As

suggested and as detailed in the following responses to specific comments, we have improved our comparison of organosulfate abundance and composition in urban Atlanta to rural Centreville. New insights to the importance of NOx in the urban environment are gained from this comparison, although we are cautious not to extrapolate our findings to other regions. Instead, we encourage future studies to determine the consistency of these findings at other urban rural pairs.

Anonymous Referee # 1, Specific comments on Abstract (page 1 line 26): "I suggest to briefly explain the formation mechanisms of organosulfate, as this knowledge is needed to understand your point here. The abstract would also benefit from an introduction in the first sentences to put the work into context for the general reader."

Authors' response to anonymous referee # 1, specific comments on abstract (page 1 line 26): We thank the referee for this suggestion and have added the following text to the abstract at page 1, line 1: "Organosulfates are secondary organic aerosol (SOA) products that form from reactions of volatile organic compounds (VOC), such as isoprene, in the presence of sulfate that is primarily emitted by fossil fuel combustion."

At line 26, we have revised the text to read: ". The greatest enhancement in concentration was observed for 2-methylglyceric acid sulfate whose formation is enhanced in the presence of nitrogen oxides (NO and NO2; NOx) and is a tracer for isoprene high-NOx SOA."

Anonymous Referee # 1, Specific Comments on Introduction: "The introduction needs a more thorough presentation of state of the art and previous studies of organosulfates in aerosols."

Authors' response to anonymous referee # 1, specific comments on introduction: We thank the reviewer for this suggestion and have provided additional discussion as detailed in our response to referee #1's specific comments on introduction.

Page 2 Line 2: "Suggest to rephrase to "Organosulfates are components of atmospheric secondary organic aerosol ... ".

Page 2, lines 2-3 originally read: "Atmospheric organosulfates are components of secondary organic aerosol (SOA) that contain a sulfate ester functional group."

This text has been revised to read: "Organosulfates are components of atmospheric secondary organic aerosol (SOA) that contain a sulfate ester functional group."

Page 2, line 3: ""Organosulfates have been detected all around the world". Please be more specific and include references."

Page 2, line 3 originally read: "Organosulfates have been detected all around the world and are estimated to contribute up to 5-9% of PM2.5 OA in the Southeastern US (Tolocka and Turpin, 2012)."

This text has been revised to read: "This class of compounds has been detected in ambient aerosols around the world, including rural, urban, forested, and coastal sites in the United States (US), China, and/or Europe (Hansen et al., 2014; He et al., 2014; Kristensen and Glasius, 2011; Lin et al., 2012; Stone et al., 2012; Ma et al., 2014). In the Southeastern US, organosulfates are estimated to contribute up to 5-9% of PM2.5 organic aerosol (Tolocka and Turpin, 2012)."

Page 2, line 6: "VOC are in the gasphase while sulfate radicals are in the particle phase."

Authors' response to referee's specific comment on page 2, line 6: We thank the referee for pointing this out. We have corrected this sentence in the revised manuscript as shown below.

Page 2, line 6 originally read: "Alternatively, they form by the sulfate radical-initiated oxidation of volatile organic compounds (VOC) (Nozière et al., 2010; Schindelka et al., 2013)...."

This text is revised to read: "Alternatively, they form by reaction of oxidized volatile

organic compounds (VOC) with sulfate radicals (Nozière et al., 2010; Schindelka et al., 2013)..."

Page 2, line 8: ""largely biogenic" - there have been many studies of biogenic precursors, and fewer of anthropogenic so this statement may not be correct because we have not yet investigated all precursors of organosulfates."

Authors' response to referee's specific comment on page 2, line 8: We agree with referee's comment. We have revised this text and the text at page 2, lines 15-17 as follows.

Page 2, lines 7-9 originally read: "Precursors of organosulfates are largely biogenic VOC such as isoprene, monoterpenes, sesquiterpenes, 2-methyl-3-butene-2-ol (MBO), and green leaf volatiles (Zhang et al., 2012; Surratt et al., 2008; Chan et al., 2011; linuma et al., 2009; Shalamzari et al., 2014)."

This text is revised to read: "Biogenic VOC precursors of organosulfates include isoprene, monoterpenes, sesquiterpenes, 2-methyl-3-butene-2-ol (MBO), and green leaf volatiles (Zhang et al., 2012; Surratt et al., 2008; Chan et al., 2011; linuma et al., 2009; Shalamzari et al., 2014)."

page 2, lines 15-17 originally read: "Since fossil fuel combustion is the major source of sulfate aerosols in the atmosphere (Wuebbles and Jain, 2001; Hidy et al., 2014; Carlton et al., 2010), organosulfates are tracers of anthropogenically influenced biogenic SOA (Hettiyadura et al., 2018)."

This text is revised to read: "Since fossil fuel combustion is the major source of sulfate aerosols in the atmosphere (Wuebbles and Jain, 2001; Hidy et al., 2014; Carlton et al., 2010), biogenic VOC derived organosulfates are useful as tracers of anthropogenically influenced biogenic SOA (Hettiyadura et al., 2018)."

Page 2, line 14: "The influence of anthropogenic VOC has been established in e.g. Asian studies."

Authors' response to referee's specific comment on page 2, line 14: We thank the referee for indicating this. We have removed this phrase from the text below.

Page 2, line 12-14 originally read: "Organosulfates have also been observed in diesel and biodiesel emissions (Blair et al., 2017) and in SOA produced from anthropogenic VOC (i.e. naphthalene, methylnaphthalene) (Riva et al., 2015) and long chain n-alkanes (Riva et al., 2016a), although the significance of these sources to ambient organosulfates has not yet been established."

This text has been revised to read: "Organosulfates have also been detected among the SOA generated from diesel and biodiesel fuel emissions (Blair et al., 2017) and in SOA produced from aromatic VOC such as naphthalene and methylnaphthalene (Riva et al., 2015) as well as long chain n-alkanes (Riva et al., 2016a)."

Anonymous Referee # 1, Specific comments on Experimental Section (page 4, line 30): "What do you mean by "30% of the measurement"? Of the value?"

Authors' response to anonymous referee # 1, specific comments on experimental section (page 4, line 30): For the organosulfates that did not have authentic standards, the analytical uncertainty was estimated as 30% of their concentration value. We have revised the text to make it clear as shown below.

Page 4, line 30 originally read: "For methyltetrol sulfate and the other organosulfates that did not have standards, 30% of the measurement was used as an estimate of the analytical uncertainty (Hettiyadura et al., 2018)."

This text has been revised to read: "For methyltetrol sulfate and the other organosulfates that did not have authentic standards, the analytical uncertainty was estimated as 30% of their concentration values (Hettiyadura et al., 2018)."

Anonymous Referee # 1 General Comments on Results and Discussion: "All headlines include the words "in Atlanta". I suggest to leave this out and only specify the comparison with Centreville in that specific headline."

Authors' response to anonymous referee # 1, general comment on results and discussion: We thank the referee for this suggestion. We have revised the sub-headings of the manuscript accordingly.

Figure 2: "The time series in Figure 2 would benefit from showing time series of relevant pollution indicators."

Authors' response to anonymous referee # 1, specific comment on Figure 2: We thank the reviewer for the suggestion, however we prefer to maintain Figure 2 as a presentation of the organosulfate measurements that were made in this study. Other pollution indicators, such as NOx were not measured at the Georgia Tech sampling site and instead were drawn from measurement archives at a nearby monitoring station. Further, NOx and sulfate only weakly correlate with the organosulfates shown in Figure 2 (Table S4 and Sect. 3.8, page 11, lines 1-5), which is consistent with their consistently high levels in urban Atlanta, GA. Therefore, we respectfully disagree that Figure 2 would benefit from additional data.

Page 6, line 14: ""many species" - can you be more specific e.g. "many other organosulfate species originating from xxx""

The text in page 6, line 14 originally read: "These results indicate that organosulfates in Atlanta during August 2015 were dominated by methyltetrol sulfate, with minor contributions from many species."

This text has been revised to read: "These results indicate that organosulfates in Atlanta during August 2015 were dominated by methyltetrol sulfate, with minor contributions from many other organosulfate species derived from isoprene, monoterpenes, and anthropogenic sources."

Page 6, line 19-20: "Please describe more clearly what you mean by these relative intensities (in the mass spectra, I assume?) of organosulfate and sulphonate species."

Page 6, line 19-20 originally read: "Major species were defined in one of two ways: 1)

having a minimum relative intensity ( $\geq 1.0\%$  for m/z 97, >12% for m/z 96, >5% for m/z 81, and >3% for m/z 80 in any of the three samples) or..."

This text has been revised to read: "Major organosulfur compounds were defined in one of two ways: 1) having a minimum relative intensity in the MS/MS spectra ( $\geq$ 1.0% for precursors to m/z 97, >12% for m/z 96, >5% for m/z 81, and >3% for m/z 80 in any of the three samples) or..."

Page 7, line 21: "As all readers might not be familiar with HILIC, I suggest to explain this reasoning in more details."

Authors' response to referee's specific comment on page 7, line 21: As suggested by the reviewer, the HILIC method used in this study is further described at page 4 line 11 and reasoning has been further explained at page 6, lines 20-22.

The text at page 7, line 21 originally read: "The organosulfate with m/z 274 has multiple isomers, while only the two isomers retaining greater than 4 min are considered to be major ones (Sect. 3.2 and Fig. 30)."

This text has been revised to read: "The organosulfate with m/z 274 has multiple isomers, while only the two isomers retaining greater than 4 min are considered to be major ones as described in section 3.2 (Fig. 3o)."

Following text is added to page 4 line 11: "Briefly, the acetonitrile rich mobile phase was held at 100% from 0 to 2 minutes, and then decreased to 85% from 2 to 4 minutes and held constant at 85% until 11 minutes."

Text at page 6, lines 20-22 originally read: "...or 2) by retaining more than four minutes on the HILIC column that results in a lower relative response due to changing mobile phase conditions, despite high ambient concentrations (Hettiyadura et al., 2017)."

This text has been revised to read: "...or 2) by retaining more than four minutes. Despite the observation that organosulfates eluting after four minutes often have higher concentrations than early-eluting species, their MS response is observed to be lower

because of the increased water content of the mobile phase (20%) that does not desolvate as efficiently as acetonitrile in the ESI source (Hettiyadura et al., 2017).

Page 7: "The first paragraph contains a lot of information and I suggest to divide into at least two paragraphs to improve readability."

Authors' response to referee's specific comment on page 7: According to the referee's suggestion we have divided the first paragraph on page 7 into two paragraphs:

"The strongest organosulfate signals observed in m/z 97, 80, and 81 precursor ion scans are associated with isoprene (Figure 1 and Table 1). Methyltetrol sulfate (m/z 215), the most abundant organosulfate observed, is produced from the acid catalyzed nucleophilic addition of sulfate to IEPOX ring (Surratt et al., 2010). Organosulfates with m/z 211 (hydroxy-methyl-tetrahydrofuranone sulfates) and 213 (dihydroxy-methyltetrahydrofuranyl sulfates), of the next-highest abundance, have been observed during photo-oxidation of isoprene (Surratt et al., 2008) and are suggested to derive from oxidation of primary alcohols in methyltetrol sulfates (Hettiyadura et al., 2015). In addition, 14 other major organosulfates identified are known to derive from isoprene and isoprene oxidation products (Table 1). Many of these organosulfates have also been identified as SOA products from diesel and biodiesel fuel emissions (e.g., 2-methylglyceric acid sulfate, lactic acid sulfate, hydroxyacetone sulfate, m/z 167, 183, 197, 211, 213, 237, 239, and 253) (Blair et al., 2017), monoterpenes (m/z 239 and 253) (Surratt et al., 2008), and/or MBO (199; C5H11SO6-) (Zhang et al., 2012). However, their moderate to strong correlations with methyltetrol sulfate (Table S1) and 2-methyltetrols (Table S2) suggest that they are mainly derived from isoprene.

Among the major organosulfate signals are those associated with isoprene oxidation under high-NOx conditions such as 2-methylglyceric acid sulfate, m/z 260 and 274. 2-Methylglyceric acid sulfate is a tracer for isoprene high-NOx SOA that is formed by the acid-catalyzed nucleophilic addition of sulfate to methacrylic acid epoxide (MAE) and/or hydroxymethyl-methyl- $\alpha$ -lactone (HMML) (Lin et al., 2013). The organosulfate with m/z 260 is a nitrooxy-organosulfate that derives from photooxidation of isoprene under high-NOx conditions (Surratt et al., 2008; Gómez-González et al., 2008). Two isomers of m/z 260 were identified in this study, while up to four isomers of m/z 260 were reported in Centreville (Surratt et al., 2008). The m/z 260 also correlated moderately with methyltetrol sulfate (r=0.539, p-value=0.005, Table S1), supporting its formation from isoprene. The organosulfate with m/z 274 is also a nitrooxy organosulfate that is derived from isoprene photooxidation under high-NOx conditions (Nestorowicz et al., in review). The organosulfate with m/z 274 has multiple isomers, while only the two isomers retaining greater than 4 min are considered to be major ones (Sect. 3.2 and Fig. 30). Their longer retention times (5.6 and 5.8 min), three additional oxygen atoms, and one unit of unsaturation suggest the presence of a carboxylic acid functional group and a hydroxyl group. Plausible structures for these two organosulfates are diastereomers of 2-carboxy-3-hydroxy-4-(nitrooxy)butan-2-yl sulfate (Table 1), which could form by the oxidation of a primary hydroxyl group in 1,3-dihydroxy-2-methyl-4-(nitrooxy)butan-2-yl sulfate (an isomer of m/z 260, C5H10SO9-, proposed by Darer et al. (2011)) to a carboxylic acid. The strong correlation of these two signals at m/z 274 with the lessoxidized isoprene nitrooxy-organosulfate (m/z 260) (r=0.860, p-value<0.001, Table S1) supports this prediction. Overall, these results indicate that isoprene is the major precursor of the most abundant organosulfates observed in this study."

Page 7, line 2: "Which "signal" do you refer to?"

Authors' response to referee's specific comment on page 7, line 2: We thank the referee for indicating this. The majority of the strongest organosulfate signals observed in m/z 97, 80, and 81 are attributed to isoprene. We have added this to the revised manuscript as shown in the authors' response to referee's specific comment on page 7.

Page 7, lines 4-5: "Please check that all compounds names are written correctly - it seems that spaces are missing in some names."

Authors' response to referee's specific comment on page 7, lines 4-5: We have revised

the nomenclature of m/z 213 and 211 according to Hettiyadura et al. (2015) as shown in the authors' response to referee's specific comment on page 7.

Page 7, line 8: "Suggest to rephrase to "have also been identified as.."

Authors' response to referee's specific comment on page 7, line 8: We have revised this sentence according to referee's comment in the authors' response to referee's specific comment on page 7.

Page 7, line 16: nitooxy -> nitrooxy

Authors' response to referee's specific comment on page 7, line 8: We thank the referee for pointing out this typo. We have corrected it in the revised manuscript as shown in the authors' response to referee's specific comment on page 7.

Page 7, line 30: "Please explain how "isoprene-derived organic aerosol" was identified and quantified. Using AMS data?"

Authors' response to referee's specific comment on page 7, line 30: IEPOX-OA was determined using a multilinear engine (ME-2) by analyzing PM1 OA mass spectra collected using ACSM. We have added this to the text in the revised manuscript as shown below.

The text at page 7, line 30 originally read: "IEPOX-derived OA accounted for 29% (3.3  $\mu$ g m-3) of PM1 OA at the nearby JST monitoring site in summer 2014, while the measured IEPOX-OA tracers (2-methyltetrols, C5-alkene triols, and 3-methyl-hydrofuran-3,4-diols) in PM2.5 (averaged 391.7 ng m-3) accounted for 3% of PM1 OA (Rattanavaraha et al., 2017), assuming negligible differences between PM1 and PM2.5."

This text has been revised to read: "By factor analysis of aerosol chemical speciation data (using multilinear engine [ME-2]), IEPOX-derived OA was estimated to account for 29% (3.3  $\mu$ g m-3) of PM1 (submicron particulate matter) OA at the nearby JST monitoring site in summer 2014, while the IEPOX-OA tracers measured in PM2.5 (2-methyltetrols, C5-alkene triols, and 3-methyl-hydrofuran-3,4-diols) accounted for 3% of

PM1 OA (Rattanavaraha et al., 2017), assuming negligible differences between PM1 and PM2.5.  $\H$

Page 8, line 32: "The dodecyl sulfate from surfactants is not covered well by the title of the section."

Authors' response to referee's specific comment on page 8, line 32: We thank the referee for indicating this error. We have revised this subheading as "Organosulfates derived from anthropogenic sources"

Page 9, line 24: "It is not so clear which organosulfates are detected for the first time ever and which are detected for the first time in Atlanta."

Authors' response to referee's specific comment on page 9, line 24: We recognize that a clarification is needed for this sub-heading. This section discusses the organosulfates that has been identified in Atlanta, but have not been detected in laboratory studies. The focus of this sub section is to give insights to their structures based on their elemental composition and retention times, and precursors based on their correlations with other organosulfates. We have revised the sub-heading of this section as "Additional organosulfates observed in ambient aerosol" to match with its contents. Also, the first paragraph of this section has been revised accordingly.

Text at page 9, line 25-30 originally read: "Three major organosulfates observed at m/z 155 (C3H7SO4-), 165, and 242 have not been previously reported in laboratory smog chamber experiments. They were detected in the PM2.5 collected from Centreville during summer 2013 (Hettiyadura et al., 2017). The m/z 155 in Atlanta correlated with most of the isoprene-derived organosulfates (Table S1), suggesting that it was derived from isoprene. Insight into the chemical composition, structure, and origins of m/z 165 and 242 are presented in the following paragraphs."

This text has been revised to read: "Three organosulfates that have not been previously reported in laboratory smog chamber experiments were detected among the

major organosulfate signals: m/z 155 (C3H7SO4-), 165, and 242. These signals were previously detected in PM2.5 in Centreville, AL (Hettiyadura et al., 2017), while new insights to their possible precursors and structures are gained here. The species with m/z 155 was previously identified as a mono-hydroxy propyl sulfate (Hettiyadura et al., 2017); in Atlanta, it correlated with most of the isoprene-derived organosulfates (Table S1), suggesting that it was derived from isoprene."

Page 10, lines 15-16: "This would need additional laboratory studies to confirm."

Authors' response to referee's specific comment on page 10, lines 15-16: We agree with the referee's comment. We have revised this text to include this as shown below.

Text at page 10, line 15-16 originally read: "As this nitrooxy organosulfate is expected to derive from m/z 260, it may useful as a tracer for isoprene SOA formed under high-NOx conditions and provide insights into atmospheric aging of isoprene-derived SOA."

This text has been revised to read: "As m/z 242 nitrooxy organosulfate is expected to derive from m/z 260, it may provide insight to the atmospheric aging of isoprene-derived SOA, although further evaluation is needed."

Page 10, Section 3.8: "Please make it more clear that you are comparing samples collected in different years. Differences in levels are thus heavily affected by differences in meteorology affecting isoprene emissions, transport and mixing of biogenic and anthropogenic pollutants."

Authors' response to referee's specific comment on page 10, 3.8: Following text is revised to clearly indicate the different years of sample collection at the two sites and its consequences.

Text at page 10, lines18-20 originally read: "To better understand the extent of which anthropogenic pollutants influence biogenic SOA formation in urban Atlanta during summer, the concentrations of the major organosulfates were compared to those measured in rural Centreville, AL during summer 2013 analyzed by similar methodology

(Hettiyadura et al., 2017; Hidy et al., 2014)."

This text has been revised to read: "To better understand the extent of which anthropogenic pollutants influence biogenic SOA formation in urban Atlanta during August 2015, the concentrations of the major organosulfates were compared with those measured in rural Centreville, AL during summer 2013 by similar methodology (Hettiyadura et al., 2017; Hidy et al., 2014).

Text at page 10, lines 23-25 originally read: "Since the absolute concentrations of these organosulfates can vary with time due to changes in meteorology and other factors influencing their formation, their relative contributions to PM2.5 OC were compared across the two sites (Table 2)."

This text has been revised to read: "Since the absolute concentrations of these organosulfates vary with time due to changes in meteorology, which affects isoprene emissions, transport and mixing of biogenic and anthropogenic pollutants, their relative contributions to PM2.5 OC were compared across the two sites (Table 2)."

Page 11, lines 1-5: "I suggest to specify that you measure inorganic sulfate as opposed to total sulfate measured by AMS."

Text at page 11, line 1-2 originally read: "The correlations of sulfate with most of the organosulfates were weak or negligible in Atlanta (Table S4), but were moderate to strong in Centreville (r = 0.5-0.8) (Table S6 in Hettiyadura et al. (2018))."

This text has been revised to read: "The correlations of inorganic sulfate with most of the organosulfates were weak or negligible in Atlanta (Table S4), but were moderate to strong in Centreville (r = 0.5-0.8) (Table S6 in Hettiyadura et al. (2018))."

Page 11, line 17: "Can other species or photochemical processes related to NOx be ruled out?"

Authors' response to referee's specific comment on page 11, line 17: No, other processes of isoprene-derived organosulfate formation such as by ozonolysis cannot be

ruled out. We have added this to the page 11, line 17 as indicated below.

Following text has been added to page 11, line 17: "However, organosulfate formation from ozonolysis cannot be ruled out (Riva et al., 2016b)."

Page 11, line 19: "I suggest to compare your results with some of the previous results obtained during SOAS."

The text at page 11, line 18-19 originally read: "Together, these results suggest a greater influence of NOx on isoprene-SOA formation in Atlanta compared to Centreville in summer."

This text has been revised to read: "While these findings are consistent with other studies that indicate a substantial influence of anthropogenic SO2 and NOx on biogenic SOA formation in the Southeastern US during summer (Rattanavaraha et al., 2016; Xu et al., 2015a), this study provides evidence for a greater influence of NOx on isoprene-SOA formation in urban Atlanta, GA compared to rural Centreville, AL in summer."

Page 11, line 22: "Do you mean standard synthesis (instead of development)?"

Authors' response to referee's specific comment on page 11, line 22: Yes, we have corrected this in the revised text as indicated below.

Text at page 11, line 21-22 originally read: "This study provides insights to the major organosulfate species that should be targets for future measurements and standard development."

This text has been revised to read: "This study provides insights to the major organosulfate species that should be targets for future measurements and standard synthesis."

Page 11, line 23: "Suggestion: "Of these, a standard for methyltetrol sulfate was synthesized" -> "Of these, only a standard for methyltetrol sulfate was previously synthesized"."

Authors' response to referee's specific comment on page 11, line 23: We thank the

referee for his/her suggestion. We have revised this text accordingly as shown below.

Text at page 11, line 23-24 originally read: "Of these, a standard for methyltetrol sulfate was synthesized (Budisulistiorini et al., 2015; Bondy et al., 2018)."

This text has been revised to read: "Of these, only a standard for methyltetrol sulfate was previously synthesized (Budisulistiorini et al., 2015; Bondy et al., 2018)."

Page 12, line 15: "I suggest to divide this into two sentences to indicate more clearly which data are in the SI and which are being published elsewhere."

Authors' response to referee's specific comment on page 12, line 15: We have divided this sentence into two, according to the referee's suggestion as shown below.

Text at page 12, line 15 originally read: "Organosulfate measurements are given in Table S5 and other PM2.5 measurements used in this study will be published soon elsewhere (Al-Naiema et al., in preparation)."

This text has been revised to read: "Organosulfate measurements are given in Table S5. Other PM2.5 measurements such as OC, inorganic sulfate, and SOA measured using GC-MS are provided elsewhere (Al-Naiema et al., in preparation)."

Table 1: "Please check that the resolution of molecular structures is adequate."

Authors' response to referee's specific comment on Table 1: We thank the referee for indicating this. We have replaced the structures in Table 1 with high resolution images as shown below.

Table 2: "The number of significant figures should indicate the uncertainty for each number and does not have to be the same for all levels of concentrations. The number of significant figures should thus be reduced for the highest concentrations."

Authors' response to referee's specific comment on Table 2: We have revised the significant figures in both Tables 1 and 2 according to the referee's comment.

**Works Cited**

[revised manuscript text omitted]

---

## Author Comment (AC2) · 2 Jan 2019

Anonymous Referee # 2, General Comments: "The study examines the influence of anthropogenic emissions to formation of organosulfates in an urban site in the Southeastern United States (SE-US). This is achieved by quantifying the major organosulfates from ambient measurements and comparing the identified compounds with those from laboratory experiment and a rural site in the SE-US, and other tracers measurements. The study reveals an enhanced formation of isoprene-derived organosulfate concentration, particularly the 2-methylglyceric acid sulfate which is known to be a tracer for high-NOx isoprene-derived secondary organic aerosol (SOA) formation mechanism.

The general objective is clear and the methods are well executed. There are some minor errors in the manuscript and some clarifications needed. Overall, I recommend accepting this manuscript for publication after corrections as detailed in the following."

Authors' response to anonymous referee # 2, general comments: We thank the referee for his/her careful review of our manuscript and for correcting the technical errors. We have made corrections to the manuscript according to the referee's comments in a point by point form as shown below.

Anonymous Referee # 2, Technical Comments 1: "Define abbreviations at their first appearance when you are using them repeatedly, such as: GA (Pg 1 Ln 12), AL (Pg 1 Ln 22)"

Page 1, lines 11-12 originally read: "This study examines the anthropogenic influence on biogenic organosulfate formation at an urban site in Atlanta, GA in the Southeastern United States."

This text has been revised to read: "This study examines the anthropogenic influence on biogenic organosulfate formation at an urban site in Atlanta, Georgia (GA) in the Southeastern United States (US)."

Page 1, lines 21-22 originally read: "Organosulfate species and concentrations in Atlanta were compared to those in a rural forested site in Centreville, AL during summer 2013, which were also dominated by isoprene-derived organosulfates."

This text has been revised to read: "Organosulfate species and concentrations in Atlanta were compared to those in a rural forested site in Centreville, Alabama (AL) during summer 2013, which were also dominated by isoprene-derived organosulfates."

Anonymous Referee # 2, Technical Comments 2: "Pg 1 Ln 11: add (US) after Southeastern United States"

Authors' response to anonymous referee # 2, technical comments 2: This technical error has been corrected in the respond to anonymous referee #2, technical comment

1 at page 1, line 11.

Anonymous Referee # 2, Technical Comments 3: "Pg 7 Lns 12-15: It is said here that the correlation between 2-methylglyceric acid sulfate with 2-methylglyceric acid is significant at r = 0.608. Based on description in Section 2.7, this correlation values is classified to be moderate. The use of "significant" is ambiguous, as it may be inferred as "strong". I recommend being consistent with the classification and description throughout the text."

Authors' response to anonymous referee # 2, technical comments 3: We agree with the referee's comment on using a consistent terminology to indicate the strengths of the correlations. However, this sentence is no longer in the revised text as it has been removed in the authors response to referee #1, specific comment on page 7.

Anonymous Referee # 2, Technical Comments 4: "Pg 11 Ln 30: insert "an authentic" before standard development."

Page 11, lines 28- 30 originally reads: "Given the ubiquity and high abundance of m/z 211 and 213 in the Southeastern US and other locations (Hettiyadura et al., 2017; Spolnik et al., 2018), they should be the next highest priorities for standard development."

This text is revised to read: "Given the ubiquity and high abundance of m/z 211 and 213 in the Southeastern US and other locations (Hettiyadura et al., 2017; Spolnik et al., 2018), they should be the next highest priorities for authentic standard development."

Works Cited

Hettiyadura, A. P. S., Jayarathne, T., Baumann, K., Goldstein, A. H., de Gouw, J. A., Koss, A., Keutsch, F. N., Skog, K., and Stone, E. A.: Qualitative and quantitative analysis of atmospheric organosulfates in Centreville, Alabama, Atmos. Chem. Phys., 17, 1343-1359, 10.5194/acp-17-1343-2017, 2017.

Spolnik, G., Wach, P., Rudzinski, K. J., Skotak, K., Danikiewicz, W., and Szmigielski,

R.: Improved UHPLC-MS/MS methods for analysis of isoprene-derived organosulfates, Anal. chem., 90, 3416−3423, 2018.

---

## Author Response (AR2)

**Journal:** ACP
**Title:** Organosulfates in Atlanta, Georgia: Anthropogenic influences on biogenic secondary organic aerosol formation
**Author(s):** Hettiyadura et al.
**MS No.:** acp-2018-834
**MS Type:** Research article

**Editor's Comments to the Author (Main text):** *The authors have reasonably addressed the comments of the two anonymous referees and they have modified their manuscript accordingly. However, the comments given below should be addressed and several alterations are needed for the Main text and Supplement before the manuscript can be published in ACP.*

*Main text:*
*Page 4, line 23: Replace ", other" by "; for other".*
*Page 4, line 24: Replace "and those" by "and for those".*
*Page 4, line 26: Replace "Organosulfates" by "For organosulfates".*
*Page 4, line 29: Replace "were calculated" by "was calculated".*
*Page 5, line 14: Replace "sulfur- containing" by "sulfur-containing".*
*Page 5, line 19: Replace "and elemental" by "and an elemental".*
*Page 5, line 23: Replace "were measured" by "was measured".*
*Page 6, line 12: Replace "are summarized" by "is summarized".*
*Page 6, line 13: Replace "select species" by "selected species".*
*Page 6, line 22: Replace "is shown" by "are shown".*
*Page 6, line 28: Replace "than m/z" by "than to m/z".*
*Page 7, line 24: "Nestorowicz et al." was published on 20 December 2018. Therefore, it should be updated here and on page 16 in the Reference list. Note also that the author list should be updated in that list.*
*Page 8, line 26: Replace "an SOA" by "a SOA".*
*Page 9, line 11: Reference is made here to Fig. S3, but this Figure does not exist. Should "Fig. S3" perhaps be replaced by "Table S3"?*
*Page 12, line 22: Reference is made here to Table S5, but this Table does not exist.*
*Page 14, lines 22-27: "Hansen, A. M. K., et al., 2014" should come before "Hansen, D. A., et al., 2003".*
*Page 17, line 6: Replace "Anal. chem." by "Anal. Chem.".*
*Page 18, line 13: Replace "in the Atmosphere" by "in the atmosphere".*
*Page 24, line 7: Replace "Nestorowicz et al. (in review)" by "Nestorowicz et al. (2018)".*

**Response to Editor's Comments (Main text):** We thank the editor for their suggestions to improve the clarity of the manuscript and to update a cited reference.  We have made all of the suggested changes, which are noted as track changes in the following version of the manuscript.

**Editor's Comments to the Author (Supplement):** *Tables S2 and S4, heading, line 1: It says "n=26" here, but there are 32 chemical formulas listed in the Tables.*

**Response to Editor's Comments (Supplement):** Regarding the suggestion for the supplement, "(n=26)" refers to the number of data points used in the correlation analysis, rather than the number of correlations investigated.

[revised manuscript text omitted]